# The Case for Evidence-Based Outdoor Recreation Interventions for Girls: Helping Girls "Find Their Voice" in the Outdoors

**Kate Evans [1], Kellie Walters [2,\*] and Denise Anderson [3]**

[1] Department of Recreation Management and Therapeutic Recreation, University of Wisconsin-La Crosse, La Crosse, WI 54601, USA; kevans@uwlax.edu

[2] Department of Kinesiology, California State University, Long Beach, CA 90840, USA

[3] College of Behavioral, Social, and Health Sciences, Clemson University, Clemson, SC 29634, USA; dander2@clemson.edu

\* Correspondence: kellie.walters@csulb.edu; Tel.: +1-916-300-0307

**Abstract:** Females' participation in outdoor recreation is often limited for a variety of reasons including social gender norms, a lack of exposure, and fear. Research has uncovered a wide range of positive outcomes for those females who do participate ranging from enhanced self-esteem and confidence to improved body image, indicating the importance of opening the outdoors as a welcoming place for all to experience. Finding Your Voice is a recreation intervention created with the focus of introducing middle school girls to outdoor recreation to increase the participants' self-efficacy and self-empowerment. Empirical research focusing on participant experiences has demonstrated promising results and the best practices from Finding Your Voice and the broader research on female empowerment in the outdoors are presented.

**Keywords:** outdoor recreation; female empowerment; single-gender; adolescent programming; adolescent girls; outdoor camp; girls' camp; youth development

---

## 1. Introduction

Research has consistently indicated that females are more likely than males to avoid or be deterred from participating in outdoor recreation because of gender-normative expectations, a lack of exposure, a lack of role models, and fear [1,2]. The outdoors has traditionally been viewed as the domain of males where females are out-of-place and unwelcome [1–3]. Gender normative socialization (e.g., "Little Red Riding Hood") teaches girls that the outdoors is a scary place they should avoid and engenders the sense that girls should be fearful while in this setting. This fear includes physical and psychological safety and is amplified by the reality that outdoor recreation normally occurs in secluded areas far from other people and potentially help if an accident, injury, or negative interaction should occur. Due to the potentially secluded nature of the outdoors, females tend to fear physically violent/harassing behaviors, crime, and the potential for accidents, injuries, or simply getting lost [4–7]. Additionally, outdoor recreation pursuits tend to have a higher perceived level of risk associated with them than other recreation activities. Research points to the socialization girls receive that leaves them more highly risk-averse than males which present an additional constraint for females to outdoor recreation [8,9].

While females tend to face more constraints than males in pursuing outdoor recreation, research has consistently concluded that when females participate in outdoor recreation, they experience a range of benefits. Those who pursue outdoor recreation demonstrate higher levels of self-esteem, self-trust, self-worth, assertiveness, self-sufficiency, independence, confidence, empowerment, and body image

as well as a greater sense of community and stress relief through interaction with the natural environment [3,10–12]. The benefits outdoor recreation participation can offer to females paints a clear picture of the importance of not only exposing girls to the outdoors but in helping them build the skills to experience sustained participation in outdoor recreation. Finding Your Voice (FYV) is an intervention for pre-adolescent girls intentionally designed to achieve this goal.

*The Intervention*

First developed in 2006, FYV is a weekend residential camp program based in Clemson, South Carolina aimed at introducing middle school girls to outdoor adventure recreation with target outcomes including increasing participants' self-efficacy and self-empowerment. In the initial planning stages of the camp, activist Maggie Kuhn's famous quote " . . . Speak your mind- even if your voice shakes. When you least expect it, someone may actually listen to what you have to say," [13] struck the co-creators of the camp (Clemson University faculty and graduate students) as encapsulating the soul of what the camp was aimed at achieving. Thus, the camp was named Finding Your Voice to evoke a sense of empowerment and personal confidence that is central to the goals of the intervention and plays on Kuhn's powerful sentiment. In its first iteration, the participants were exposed to a range of activities including rock climbing, backpacking, yoga, rugby, and field hockey. In addition, campers participated in educational sessions on topics including nutrition, body image, self-esteem, college life, and STEM careers. FYV was created based on the theoretical framework of self-efficacy theory (i.e., Bandura's Self-Efficacy Theory) which drove the creation of program details [14].

Self-efficacy, or one's belief in their competency at a task, is grounded in the expectations one has for how well they will perform that task. Based on Bandura's work, there are four main sources through which individuals build these efficacy expectations: performance accomplishments (successful experiences), vicarious experience (observing others similar to oneself succeed), verbal persuasion (being told one can succeed), and emotional/physiological arousal (emotional/physiological responses in a given situation that provide either positive or negative feedback to an individual) [14]. The theory also details that an individual's efficacy expectations are a "major determinant" of how much effort they are willing to put into the task, and how well they will persevere in that task [14]. Empirical research has demonstrated the effectiveness of interventions aimed at enhancing self-efficacy in physical activity [15,16] including long and short-term programs [17–20] aimed at females [20,21] and youth [22–24], and those focused on actual behavioral change [25].

Because of its roots in self-efficacy theory, FYV was intentionally crafted to maximize the opportunity for participants to enhance their self-efficacy. First, all of the counselors, as many of the instructors as possible, and all of the campers are females of a similar age for their respective groups. This provides models for the participants to view interacting in the camp environment and the various activities positively and successfully, thereby allowing them to see others, similar to themselves, succeeding. In addition, the activities are focused on basic, entry-level skills to heighten the opportunity for the campers to be successful in trying an activity, many for the first time. Further, the counselors are trained prior to each camp on the basic tenets of self-efficacy theory to ensure they understand the importance of encouraging participants to view the success of others, of providing realistic positive verbal feedback, of working with each participant to help them succeed in each activity (e.g., providing positive reinforcement and helpful instructions during the activity), and to properly handle discussions around the anxiety, stress, or fear campers might feel prior to or during activities.

FYV has not remained static, however. Over time, new research, theory, and a continued goal of focusing the camp on its central focus—outdoor adventure recreation—has resulted in a wide range of changes to the program. Based on more current research on female participation in outdoor adventure recreation, the program was tweaked in recognition of these new insights. First, a continued focus was placed on developing resiliency skills in the campers (enhanced self-efficacy, self-esteem, self-empowerment) with a heightened understanding of how important this factor can be in successful outdoor recreation participation [26]. In addition, more emphasis was placed on inserting the camp

into the natural environment after research identified how important this connection to the outdoors can be in overcoming constraints over time in outdoor recreation [27]. This included the transition of all activities to an outdoor facility and away from on-campus facilities, a schedule change to ensure that all campers would participate in kayaking, backpacking, and rock climbing—activities central to the outdoor recreation focus of the camp, and at least one meal prepared and eaten outdoors. In addition, a female athlete who has reached a high level of success in outdoor recreation is integrated into the camp (e.g., a keynote speaker, highlight videos, interviews) as research has indicated the importance of connecting young participants to mentors and exposing them to skilled female athletes [26]. Through its evolution, FYV has stayed focused on its original goal of introducing the outdoors as a welcoming place that has served 250+ adolescent girl campers over 7 years.

## 2. Materials and Methods

Data from the FYV camp has been collected for a total of eight, nonconsecutive years (once in 2006 and from 2013–2019). The original camp in 2006 was run as a pilot camp and started again in 2013 due to personnel and funding support. In two instances, 2006 and 2013, a survey, based on a task-specific self-efficacy measurement focused on physical activity self-efficacy, a key theoretical basis for the camp itself [28] was given to the participants. Questions included related to support seeking (e.g., parental/adult support), barriers to participation (e.g., how likely participants felt they would be able to overcome constraints to participate in physical activity), and positive alternatives (e.g., how likely participants felt they were to choose physical activity over non-physically active activities). Adolescent girls between the ages of 9–13 who participated in FYV ($N = 75$) on these two instances completed a pre- and post-test survey on the first and last day of camp. Additionally, each year, participants engaged in focus groups concentrated on the key theoretical elements of camp (e.g., constraints to participation, social support, constraint negotiation, etc.) on the last day of camp. For the quantitative measure, paired sample t-tests were conducted to assess changes in self-efficacy with a $p$-value of $<0.05$ indicating significance. The focus group data were analyzed using open axial coding. This research was approved by the authors' Institutional Review Board and both parental consent and child assent were received.

## 3. Results

This research conducted on FYV has consistently supported the positive benefits related to participation in the camp. Results of the quantitative evaluation has indicated a statistically significant increase in the participants' physical activity self-efficacy (e.g., $t = 3.912$, $df = 39$, $p < 0.001$; $t = -4.225$, $df = 34$, $p < 0.001$) from pre- to post-program. Additionally, qualitative findings have uncovered themes related to various elements of the camps' design including related to confidence-building ("[After doing the activities] I felt like I could do anything. And I felt strong and powerful."), and an appreciation for the camp environment ("I felt better about myself here because there was no one that was going to be mean, like we were all gonna be accepted.") including the all-female experience ("[If boys were here], you feel pressure that you'd have to get to the top of the rock wall . . . because they can. The people that belayed, they said you don't have to get all the way up, you can just try [your best]."). Another theme that has consistently arisen in focus groups relates to the resiliency skills the campers report building:

> Going on the rock wall, I sort of fell and shifted and stuff so that was scary but it really wasn't that bad. And then kayaking I was sort of scared when I was flipping over that I would get stuck or hit in the head, but it wasn't that bad and it was actually kind of fun to flip over. And then backpacking, you would think being in the woods would be scary like maybe a bear or snake or something, but it was really fun.

Closely related to this is the way campers' perspectives change as they think towards the future and participating in similar activities again: "Now if we do these activities again like sometime during our life, I would step up and be the first person to do it rather than being the last or middle person like

today." These findings, while limited by geographical location and the particular participants who have participated in FYV and the data collection, are encouraging and point towards the importance of continuing to offer interventions to girls to enhance their exposure to the outdoors and, hopefully, their perception of that space as welcoming to all.

## 4. Discussion

### 4.1. Expanding Outdoor Interventions

While the FYV model has been presented here, it is not the only option for introducing girls to outdoor recreation. With the goal of broadening the view of the outdoors as an environment open to females, the focus should simply be on implementing more intentionally designed, evidence-based female-focused interventions. Four areas that deserve intentionality in the design are the structure and focus of the intervention, relationship building, and ongoing evaluation.

### 4.2. Structure

First, the basic structure of the program should be designed in a way to best support the female participants' comfort and goal achievement. One major consideration that should be made is the gender makeup of the participants and support staff. There is a wide range of evidence that suggests that all-female environments may best deconstruct the traditional gender roles and gender typing inherent in outdoor recreation. A single-gender environment can help participants to feel more relaxed and be free of gender stereotypes when taking on these new activities in a potentially uncomfortable environment [29–31]. A single-gender environment can also eliminate the possibility that the girls take on more traditional, submissive roles when in the activities (e.g., allowing a male to take the "control" position at the stern of a canoe) [3]. This all-female structure may allow girls to experience their own success as opposed to comparing it with male participants who are (perceived to be) more skilled and experienced. Additionally, a single-gender environment may allow participants to more easily move away from a context where one's body is seen as an "object" for others to observe [32]. Thus, the general concern of trying to impress boys in the group is removed and, as Mitten [29] concluded, females are more likely to feel unconditionally supported and accepted in all-female environments [30,31].

Additionally, research indicates all-girls programs tend to reduce competition, enhance levels of participation, and allow girls to focus on their own growth [30]. The addition of an all-female staff provides an additional layer of support for the self-efficacy and confidence of the girls in even attempting new activities. The provision of role models who are "like" the participants gives them an extra layer of messaging to support their ability to succeed [14]. It is important to note that creating a single-gender environment has its own drawbacks that must also be addressed through the design of the program. First, Whittington et al. noted the importance of creating an inclusive environment to avoid the formation of cliques and/or divisions in the group as well as creating an intentionally supportive environment [30]. Some of these design elements (including the importance of relationship building) will be discussed further. Eagleman also found that the hyper-femininity (e.g., emphasis on pink, frilly, and other stereotypically feminine markers) sometimes present in single-gender activities can be viewed negatively [33]. Thus, removing (or never implementing) these stereotypical images or design elements is important to consider. Research has also indicated that single-gender environments are not always the most appropriate or necessary makeup for creating positive experiences in outdoor recreation [26,34]. For instance, while an all-female environment may be helpful in early experiences in outdoor recreation, sustained participation in the outdoors will require mixed-gender environments [26]. Similarly, not all outdoor recreation environments are created equally, so when the need for competition or cooperation between participants is minimal, gender differences are less salient [34]. However, these findings related to the specific environments of competition climbing and the experiences of highly skilled female mountain guides. As such, even with this research in mind, for the purposes

of "introducing" girls to the outdoors, the single-gender environment as suggested here is, arguably, most conducive to creating positive outcomes for participant self-efficacy [29–32].

In addition to the general environment, inherent to the structure of the intervention must be an intentional skill or competency building focus. In the case of FYV, the main focus is on enhancing self-efficacy. As outlined in the program description, all of the basic program elements were designed around the research on how self-efficacy beliefs are built. While self-efficacy does not have to be the focus of your intervention, whatever competency you choose to focus on must be built into all facets of your program to create an environment conducive to addressing your desired outcomes. This should include, but not be limited to, the physical environment, the make-up of staff (e.g., highly skilled vs. novice), activities offered (including the level of difficulty/challenge), feedback mechanisms (e.g., how staff are trained to respond to/support participants), opportunities for participant reflection (e.g., journaling, group discussions), and your own evaluation of the program's outcomes.

### 4.3. Focus

Beyond the structure of the intervention, the focus of the intervention should be honed in on a deliberate emphasis on key aspects of outdoor recreation and the related skills research has indicated as necessary for success in the outdoors. First, previous research indicates that a passion for the outdoors is a key component of overcoming the constraints typical to females' experiences [27]. This indicates that in addition to introducing girls to the skills needed, immersing them in the outdoors is important in allowing them to connect with the environment as a component of that experience. Integrating the outdoors can be as simple as offering outdoor vs. indoor rock climbing or conducting a backpacking course in the woods rather than an urban greenspace. Additional immersion could be built-in during meals, the location of reflection activities, or the location of supporting lessons (i.e., those not related directly to outdoor skills such as resiliency-building, self-esteem, etc.). Finally, connecting your participants with resources for continued participation in the activities can be critical to their long-term, sustained involvement. Resource binders with an overview of the skills they learned at camp, listings of local providers, maps of local trail systems, or listings of other upcoming outdoor recreation opportunities can provide a roadmap for continued involvement in this early confidence-building stage of participation.

Beyond the acute focus on the outdoors and outdoor recreation, providing a broader foundation focused on the deconstruction of gender norms as well as building resiliency skills will help support the goal of breaking down barriers to outdoor recreation participation [26,30]. The FYV model includes a range of educational sessions that changes from year-to-year but always focus on the broader goal of challenging normative gender roles and arming the participants with the skills needed to be self-sufficient and confident when faced with the inevitable challenges present not only in outdoor recreation but in pursuit of any of their goals.

### 4.4. Relationship Building

A bevy of research also indicates how critical it is that relationship building is at the core of interventions created to empower females in the outdoors. First, be sure to include female role models in the activities you are introducing. While staff will certainly fill this role to an extent, it can also be empowering to expose girls to females participating at the highest levels in the activities they are learning. Providing role models will help to cultivate the perspective of the outdoors as a welcoming place for females while simultaneously demonstrating the ability for women to not only participate, but thrive and excel [2,26,35]. These role models could be introduced in the form of guest speakers or visitors to the program, but could also be introduced through highlight videos.

In addition to role models, the literature points to the importance of having connections with other participants in cultivating sustained involvement in the outdoors [26,31]. One programmatic aspect that can help to address this point is simply creating a culture and opportunities within the program focused on building relationships and connections between participants. Not only does this

create a more open and welcoming atmosphere during the program and work to prevent some of the pitfalls to single-gender programs [30], but it has the potential to create longer term relationships participants can rely on for continued participation. Next, one-off interventions can certainly be helpful, but providing longer term mentorship and support opportunities can be key in helping females navigate longer-term participation [26]. Thus, building in continued, ongoing meetings, communication, or activities involving the girls and camp staff can help to cultivate this critical social support. This could come in the form of quarterly meetings, monthly activities, or even a social media group dedicated to keeping the community connected. Finally, and closely related to these relationship supports, is to build collaboration and connection between the program and providers of outdoor recreation in the region the intervention is offered. Expanding the connection females feel beyond just those they participated with or just a single organization that initially offered the intervention continues to expand the network participants can rely on to support their ongoing pursuit of outdoor recreation.

*4.5. Evaluation*

A final and critical aspect of creating and implementing these interventions is the need for evaluation, both formative and summative. During the intervention itself, closely monitoring the program can allow you to quickly remedy any issues that are arising that could be detrimental to the experience of the girls (e.g., the formation of cliques or divisions, adjusting challenge levels). After the intervention, conduct a formal evaluation to assess how well you achieved the stated goals of the experience (e.g., increased self-efficacy, self-esteem). Understanding what the participants actually experienced compared with what the intervention was designed for them to experience can provide critical information on further refining the program. Conducting these formal assessments via both quantitative and qualitative means will allow for a simultaneous assessment of particular constructs (e.g., self-efficacy) as well as a deeper dive into the participants' experiences. It is also important to conduct a summative assessment with the staff involved in the intervention to provide a broader view of the program and its successes and shortcomings. In addition to an evaluation of the intervention, research continues to move our understanding of females' experiences in the outdoors forward and sheds light on new details and nuances that had not been considered before. Thus, in addition to intervention-specific evaluation and improvements, staying aware of new research and best practices and allowing for the evolution of programs based on each will allow for the greatest chance of success in meeting both the participants' needs and the stated program outcomes. Most importantly, these programs, grounded in recreation, must rely on key components of what recreation should be (e.g., freedom, autonomy, intrinsic motivation). To fully benefit the participants, these programs must be embedded in a leisure ethos, ripe with opportunities for autonomy and freedom for the participants, and, maybe most critically, should be fun.

## 5. Conclusions

With the current physical and emotional state of adolescent girls, adolescent girls must be introduced to ways to improve their health, including outdoor programming. Drawing on research conducted on one such intervention as well as the larger body of research focused on providing greater access to the outdoors for girls, this paper highlights the importance of including outdoor activities and socio-emotional learning as part of programming for adolescent girls. Future research should focus on ways to continue to support program participants after program completion, including the use of continued education and communication via social media, involving past participants as junior camp counselors, and promoting opportunities for future group gatherings.

**Author Contributions:** All authors contributed to the design and implementation of the research. K.E. conducted the analysis of results and initial writing of the manuscript and K.W. and D.A. assisted with revisions and edits to the manuscript. All authors have read and agreed to the published version of the manuscript.

**Funding:** This research received no external funding.

**Conflicts of Interest:** The authors declare no conflict of interest.

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
