# Peer review of "The Case for Evidence-Based Outdoor Recreation Interventions for Girls: Helping Girls “Find Their Voice” in the Outdoors"

_education, doi:10.3390/educsci10120363_

Round 1
Reviewer 1 Report
Overall, I found the article to be well-written and the argument for same-gender recreation-based programs for young girls to be convincing. The discussion of the structure and intent of the program was also presented well. As a participant in a similar program for young adult cancer survivors, the model of the program described in this paper was, in fact, very similar, especially in terms of relationship building, interventions, and focus though it was a co-ed group. I see firsthand the benefits of such a program and argue that young girls could benefit. The only component that was missing, through no fault of the author(s) is working to continue to provide support well after the experience has ended. I would like to hear more about how this would take place.
The author(s) may wish to consider research conducted by McNiel, Harris, and Fondren (2012) titled "Women and the Wild: Gender Socialization in Wilderness Recreation Advertising," published in Gender Issues, DOI: 10.1007/s12147-012-9111-1. The authors also discuss a lack of women's participation in the outdoors and considered Backpacker and Outside magazines as possible gateways for entry.
Other citations/research to consider:
- Maller, C., Townsend, M, Pryor, A., Brown, P. & St. Leger, P. (2005). Healthy nature healthy people: ‘contact with nature’ as an upstream health promotion intervention for populations. Health Promotion International, 21, 45-54.
- Manning, R. & More, T. (2002). Recreational values of public parks. The George Wright FORUM, 19, 21-30.
- Vries, S. d., Verheij, R., Groenewegen, P., & Spreeuwenberg, P. (2003). Natural environments-healthy environments? an exploratory analysis of the relationship between greenspace and health. Environment and Planning, 35, 1717-1731.
- Warren, K. (1996). Women’s outdoor adventures: myth and reality. In K. Warren (Ed.), Women’s voices in experiential education (pp. 10-17). Debuque, IA: Kendall Hunt.
- Warren, K. (2002). Preparing the next generation: social justice in outdoor leadership education and training. Journal of Experiential Education, 25, 231-239.
- Wearing, B. & Wearing, S. (1988). “All in a day's leisure”: gender and the concept of leisure. Leisure Studies, 7, 111-123.
- Whitson, D. (1994). The embodiment of gender: discipline, domination, and empowerment. In S. Birrell and C. L. Cole (Eds.), Women, sport, and culture (pp. 353-371). Champaign: Human Kinetics.
- Henderson, K. (1995). Marketing recreation and physical activity programs for females. Journal of Physical Education, Recreation, and Dance, 66, 53-57.
- Henderson, K. (1996). “Feminist Perspectives on Outdoor Leadership.” In K. Warren (Ed.), Women’s voices in experiential education (pp. 107-117). Debuque, IA: Kendall Hunt.
- Henderson, K. & Bialeschki, D. (1993). Fear as a constraint to active lifestyles for females. Journal of Physical Education, Recreation, and Dance, 64, 44-47.
- Cohen, D, McKenzie, T. L., Sehgal A, Williamson, S., Golinelli, D. & Lurie, N. (2007). Contribution of public parks to physical activity. American Journal of Public Health, 97, 509-514.
- Cole, E., Erdman, E., & Rothblum E. D. (1994). Wilderness therapy for women: the power of adventure. Binghamton: Haworth Press.
Author Response
Education Sciences: Reviewer One’s Feedback
Overall, I found the article to be well-written and the argument for same-gender recreation-based programs for young girls to be convincing. The discussion of the structure and intent of the program was also presented well. As a participant in a similar program for young adult cancer survivors, the model of the program described in this paper was, in fact, very similar, especially in terms of relationship building, interventions, and focus though it was a co-ed group. I see firsthand the benefits of such a program and argue that young girls could benefit.
The only component that was missing, through no fault of the author(s) is working to continue to provide support well after the experience has ended. I would like to hear more about how this would take place.
Thank you for this feedback. We agree that this information should be included in the paper. We added more about how we can continue to provide support to the participants in the conclusion section on page 13 (lines 289-295).
The author(s) may wish to consider research conducted by McNiel, Harris, and Fondren (2012) titled "Women and the Wild: Gender Socialization in Wilderness Recreation Advertising," published in Gender Issues, DOI: 10.1007/s12147-012-9111-1. The authors also discuss a lack of women's participation in the outdoors and considered Backpacker and Outside magazines as possible gateways for entry.
Thank you for providing these resources to us. We reviewed them all and added the following references (and useful information/context from them) below:
- Maller, C., Townsend, M, Pryor, A., Brown, P. & St. Leger, P. (2005). Healthy nature healthy people: ‘contact with nature’ as an upstream health promotion intervention for populations. Health Promotion International, 21, 45-54 on page 2 (lines 47-48).
- Whitson, D. (1994). The embodiment of gender: discipline, domination, and empowerment. In S. Birrell and C. L. Cole (Eds.), Women, sport, and culture (pp. 353-371). Champaign: Human Kinetics on page 8 (lines 170-172).
Reviewer 2 Report
Dear authors, I feel your enthusiasm, the topic is interesting, I believe the supporting girl is a very significant job in the country you live in. But, I am sorry, your article is not a research paper.
I miss:
a) the typical structure of a research article, especially methodology (!) and discussion and conclusion,
b) the research work with data (analysis, open and axial coding, making categories etc.),
c) the basic information about the intervention (place, state etc.).
I strongly disagree with your generalizations. Your statements are not generally valid. At least you can't say that from the presented data. Your work – supporting the girls in outdoor recreation – is undoubtedly significant. You need to improve your research methodology. The case study also has its rules. Your sincerely!
Author Response
Education Sciences: Reviewer 2 Comments
Dear authors, I feel your enthusiasm, the topic is interesting, I believe the supporting girl is a very significant job in the country you live in. But, I am sorry, your article is not a research paper.
I miss:
- the typical structure of a research article, especially methodology (!) and discussion and conclusion,
We appreciate your feedback. Since this was written as a case study/communication submission, the structure is different than a typical research manuscript. To make this paper read more like a traditional research paper, we added a specific methods section on pages 5 and 6 (lines 106-123), and conclusion on page 13 (lines 289-295).
- b) the research work with data (analysis, open and axial coding, making categories etc.),
See response above. We added a specific methods section on pages 5 and 6 (lines 106-123), which includes information about data analysis.
- the basic information about the intervention (place, state etc.).
We added the specific place/state on page 3 (lines 54 and 55).
Reviewer 3 Report
Thank you for this and your work supporting young women in outdoor education. Having just introduced a baby girl to the world made reading this piece particularly interesting. This paper/communication is fine, and I attached a scanned version of the PDF with my notes in the margins, hope you are able to read them. I would like to see you build out your methods a bit better, for really all I remember reading was evals and quant/qual methods. Give us a better sense of how you went about finding your findings, which I also think can be developed a bit further and embellished in a formal conclusion section, which there isn't one. Also, assuming these were minors, was there any ethical considerations to mention? Releases?

Author Response
Education Sciences: Reviewer 3 Comments
Thank you for this and your work supporting young women in outdoor education. Having just introduced a baby girl to the world made reading this piece particularly interesting. This paper/communication is fine, and I attached a scanned version of the PDF with my notes in the margins, hope you are able to read them.
I would like to see you build out your methods a bit better, for really all I remember reading was evals and quant/qual methods.
Thank you for your feedback. We agree that a methods section would improve this paper and have added a specific methods section on pages 5 and 6 (lines 106-123).
Give us a better sense of how you went about finding your findings, which I also think can be developed a bit further and embellished in a formal conclusion section, which there isn't one.
We added more about how we got our findings in the methods section (see note above) as well as in the newly added conclusion section on page 13 (lines 289-295).
Also, assuming these were minors, was there any ethical considerations to mention? Releases?
We added a statement about IRB approval and parental consent and child assent on page 6 (lines 121-123).
Lastly, we reviewed the manuscript with your handwritten comments and responded to your questions/comments with text boxes on the scanned PDF document.
Round 2
Reviewer 2 Report
Dear authors,
you made good edits and you moved the article significantly.
I now believe that it is possible to publish an article as a research article.
Thank you for completing the methodology. I still miss information on the research method in the abstract.
I repeat: I strongly disagree with your generalizations. Your statements are not generally valid. At least you can't say that from the presented data. Your experience is interesting, but such general conclusions cannot be drawn from your study. You over-generalize right from the start to he end. Your assumption that "girls have less chance to spend time in nature" is not universally valid. You should take this into account in your work. You should describe the reasons why we think so, why it is so in your country, why it is so in other countries. And that in some countries it is different. Or at least admit that it may be different in other countries.
Please take this into account in the title of the article, in the abstract (for example: Females’ participation in outdoor recreation in South Carolina is often limited), in the text of the discussion and in the Conclusion (With the current physical and emotional state of adolescent girls in South Carolina, it is imperative that adolescent girls are introduced to ways to improve their health, including outdoor programming …) etc.
I recommend changing the title of the article to better reflect the content (for example: “The Case for Evidence-Based Outdoor Recreation Interventions for Girls from South Carolina”). The information about the year (2006) in the title is irrelevant. Especially when you had a big break between years of camp implementation and research.
I'm not sure if it's helpful to state „helping Girls “Find Their Voice”. You need to explain more what Helping Girls “Find Your Voice” is? I feel the connection with Carol Gilligan's work. But I am sorry I haven't heard of this yet. I have search on the internet and I found plenty of links. Not just a connection to your camps. It seems to me that this is a movement. Did you run it? Or are you part of it? That needs to be explained. Why do you use the same name?You should explain in the text why and how you refer to this movement. You should explain in the text what it is briefly about. And you should write why you refer to this movement. Are you its founder? Or do you follow him? Girls can find their voice in other ways, right?
Similarly, you generalize the conclusions too much. Just because the girls liked your segregated camp doesn't mean they liked the segregation. You can't rule out that girls who like a coed education like the same or even more. You report on your experience with a segregated campsite. Why did you choose a segregated campsite? Surely you had reasons. And these reasons affect your interpretation of the results too. The success or failure of a segregated camp depends on many other factors. What are your schools like? Segregated or coeducated?
I ask for the wording to be modified so that it is clear that the authors are aware of the weaknesses of their research and the correction of interpretation.
I'm looking forward to the final adjustments.
Author Response
I repeat: I strongly disagree with your generalizations. Your statements are not generally valid. At least you can't say that from the presented data. Your experience is interesting, but such general conclusions cannot be drawn from your study. You over-generalize right from the start to he end. Your assumption that "girls have less chance to spend time in nature" is not universally valid. You should take this into account in your work. You should describe the reasons why we think so, why it is so in your country, why it is so in other countries. And that in some countries it is different. Or at least admit that it may be different in other countries.
Our premise that females tend to have less access to the outdoors and tend to face additional constraints not faced by males is grounded in a wide range of research outlined in the introduction. We feel this has been clearly articulated and, based on the extant research, our foundation is solid and supported. The research highlighted does include data from various countries, so the evidence points to this as a broader issue beyond even the bounds of the US.
Please take this into account in the title of the article, in the abstract (for example: Females’ participation in outdoor recreation in South Carolina is often limited), in the text of the discussion and in the Conclusion (With the current physical and emotional state of adolescent girls in South Carolina, it is imperative that adolescent girls are introduced to ways to improve their health, including outdoor programming …) etc.
As grounded in the literature review, the issue extends beyond South Carolina, so, as the manuscript discusses, the best practices outlined are based on a wider swath of research than just FYV. Thus, we feel limiting all of the description to just South Carolina does a disservice (and does not accurately reflect) the wider range of research drawn on for the manuscript both in the literature review and in the discussion section.
I recommend changing the title of the article to better reflect the content (for example: “The Case for Evidence-Based Outdoor Recreation Interventions for Girls from South Carolina”). The information about the year (2006) in the title is irrelevant. Especially when you had a big break between years of camp implementation and research.
The title has been changed to remove the date.
I'm not sure if it's helpful to state „helping Girls “Find Their Voice”. You need to explain more what Helping Girls “Find Your Voice” is? I feel the connection with Carol Gilligan's work. But I am sorry I haven't heard of this yet. I have search on the internet and I found plenty of links. Not just a connection to your camps. It seems to me that this is a movement. Did you run it? Or are you part of it? That needs to be explained. Why do you use the same name?You should explain in the text why and how you refer to this movement. You should explain in the text what it is briefly about. And you should write why you refer to this movement. Are you its founder? Or do you follow him? Girls can find their voice in other ways, right?
Additional history was included to better describe why the camp was named Finding Your Voice and how the name ties into the central focus of the camp (page 3 lines 57-63).
Similarly, you generalize the conclusions too much. Just because the girls liked your segregated camp doesn't mean they liked the segregation. You can't rule out that girls who like a coed education like the same or even more. You report on your experience with a segregated campsite. Why did you choose a segregated campsite? Surely you had reasons. And these reasons affect your interpretation of the results too. The success or failure of a segregated camp depends on many other factors. What are your schools like? Segregated or coeducated?
The language has been softened throughout to achieve this goal. Throughout our discussion, we presented research (in addition to the findings related to this intervention) that describes the specific components of all-female environments that may cultivate better experiences for females in the outdoors, especially as it relates to early exposure. In addition, we presented research that has pointed away from the need for single-sex environments. Thus, we feel readers have a full picture of the evidence and can draw their own conclusion related to the efficacy of a single-sex environment – especially when they consider the particulars of the environment in which they may be considering implementing such a program or working to achieve similar goals.
I ask for the wording to be modified so that it is clear that the authors are aware of the weaknesses of their research and the correction of interpretation.
An additional statement has been added to the results section (page7 lines 156 and 157) to acknowledge the limitations of the research. We believe this and the other changes we have made throughout the manuscript have addressed this concern.
I'm looking forward to the final adjustments.